# The Role of Reduced Methionine in Mediating the Metabolic Responses to Protein Restriction Using Different Sources of Protein

**DOI:** 10.3390/nu13082609

**Published:** 2021-07-29

**Authors:** Han Fang, Kirsten P. Stone, Sujoy Ghosh, Laura A. Forney, Thomas W. Gettys

**Affiliations:** 1Laboratory of Nutrient Sensing & Adipocyte Signaling, 6400 Perkins Road, Pennington Biomedical Research Center, Baton Rouge, LA 70808, USA; han.fang@pbrc.edu (H.F.); kirsten.stone@pbrc.edu (K.P.S.); 2Laboratory of Computational Biology, Pennington Biomedical Research Center, Baton Rouge, LA 70808, USA; Sujoy.ghosh@pbrc.edu; 3Program in Cardiovascular and Metabolic Disorders and Center for Computational Biology, Duke-NUS Medical School, Singapore 169857, Singapore; 4Department of Integrative Biology and Pharmacology, University of Texas Health Science Center at Houston, 7000 Fannin St, Houston, TX 77030, USA; Laura.A.Bobart@uth.tmc.edu

**Keywords:** essential amino acid, nutrient sensing, gene set enrichment, protein restriction, FGF21

## Abstract

Dietary protein restriction and dietary methionine restriction (MR) produce a comparable series of behavioral, physiological, biochemical, and transcriptional responses. Both dietary regimens produce a similar reduction in intake of sulfur amino acids (e.g., methionine and cystine), and both diets increase expression and release of hepatic FGF21. Given that FGF21 is an essential mediator of the metabolic phenotype produced by both diets, an important unresolved question is whether dietary protein restriction represents de facto methionine restriction. Using diets formulated from either casein or soy protein with matched reductions in sulfur amino acids, we compared the ability of the respective diets to recapitulate the metabolic phenotype produced by methionine restriction using elemental diets. Although the soy-based control diets supported faster growth compared to casein-based control diets, casein-based protein restriction and soy-based protein restriction produced comparable reductions in body weight and fat deposition, and similar increases in energy intake, energy expenditure, and water intake. In addition, the prototypical effects of dietary MR on hepatic and adipose tissue target genes were similarly regulated by casein- and soy-based protein restriction. The present findings support the feasibility of using restricted intake of diets from various protein sources to produce therapeutically effective implementation of dietary methionine restriction.

## 1. Introduction

Studies of the short-term metabolic effects of dietary protein restriction and methionine restriction (MR) have been pursued in parallel in recent years, with studies conducted in the last decade identifying significant similarities among the responses to the two dietary regimens (reviewed in [1,2,3,4,5]). Depending on the severity of dietary protein restriction, the responses can range from modest slowing of growth and improvements in metabolic health to significant growth stunting, delayed wound healing, and decline in health [6,7]. The responses to dietary MR are also concentration-dependent, where severe restriction causes rapid weight loss [8], whereas a slightly less severe restriction modestly reduces growth and produces beneficial metabolic effects [8]. In contrast, an even less severe restriction involving a 2- to 3-fold reduction in dietary methionine was without effect on growth, energy balance, or body composition [8]. Collectively, these studies illustrate that restricting dietary methionine to a specific range is highly beneficial, while a more severe restriction can be detrimental, and a less severe restriction is without effect [8].

Since the early work of Rothwell and Stock showing that dietary protein restriction reduced weight gain by increasing energy expenditure (EE) [9,10], significant effort has been devoted to identifying the underlying mechanisms for this response. Recent work has shown that the increase in EE produced by protein restriction is linked to increased release of hepatic FGF21 [11,12,13]. However, the identity of the amino acid(s) being sensed after protein restriction remains an open question. Yap et al. [5] used several approaches to identify the key amino acid(s) being limited by protein dilution, and they make a compelling case that the reductions in threonine and tryptophan are essential to several adaptive responses. A case can also be made that dietary casein restriction limits methionine to a concentration range previously shown to activate FGF21 and increase EE [8]. Additionally, although restriction of several essential amino acids also increases FGF21 and EE [14,15,16], only methionine restriction downregulates hepatic lipogenic genes and reduces hepatic lipids [17]. The goal of the present work was to test whether the well-established metabolic phenotype produced by elemental methionine restriction could be recapitulated using different protein sources to produce a comparable degree of sulfur amino acid restriction. The overall aim of this work is to develop protein-restricted therapeutically effective diets that reproduce the beneficial metabolic effects of dietary methionine restriction. Using mice fed protein-restricted diets formulated with different protein sources but with matched restrictions of sulfur amino acids, it is shown that casein protein restriction and soy protein restriction produce comparable metabolic effects on energy balance, hepatic gene expression, and adipose tissue gene expression that are consistent with the biological responses to dietary MR.

## 2. Materials and Methods

### 2.1. Diets and Animals

All experiments were approved by the Pennington Biomedical Research Center Institutional Animal Care and Use Committee on the basis of guidelines established by the National Research Council, the Animal Welfare Act, and Public Health Service Policy on the humane care and use of laboratory animals. Pelleted diets (Research Diets Inc., New Brunswick, NJ, USA) were formulated using either casein or soy isolates as the sole protein source. The respective amino acid content of the two protein sources used for diet formulation is provided in Appendix A. A control, casein-based diet (e.g., Cas-20%) was formulated containing 20% casein, and cystine was added to match the cystine content of the soy-based control diet (Table 1). A casein-based low protein diet (e.g., Cas-5%) was formulated containing 5% casein, and cystine was added to match the cystine content of the soy-based low protein diet (Table 1). The control, soy-based diet (e.g., Soy-20% + SAA) was formulated containing 20% soy, with additions of methionine and cystine to match the final concentrations of sulfur amino acids in the casein-based control diet (Table 1). The soy-based low protein diet (e.g., Soy-10%) was formulated to contain 10% soy as this amount of soy protein contained amounts of methionine and cystine that matched the content of these amino acids in the casein-based low protein diet (Table 1). Lastly, a soy-based diet containing 20% soy and no added sulfur amino acids was formulated as a negative control diet (e.g., Soy-20%). Diets were formulated as extruded pellets containing 22% fat, with a final energy content of 17.12 kJ/g (Table 1).

Fifty male C57BL/6J mice were purchased from Jackson Laboratory (Bar Harbor, ME, USA) at 9 weeks of age and quarantined for one week upon arrival. Thereafter, the mice were singly housed and 20 were put on Cas-20% control diet and 30 were put on the Soy-20% control diet. Mice were acclimated to the diets for two weeks, and at twelve weeks of age, half the mice on the Cas-20% diet were switched to the Cas-5% diet, while the other half remained on Cas-20% diet. Mice on Soy-20% diet were evenly divided into three groups, with 10 switched to the Soy-20% + SAA diet, 10 continued on the Soy-20% diet, and the remaining 10 switched to the Soy-10% diet. Food and water were provided ad libitum, and animals were housed at 23 °C and maintained on a 12-hour light/dark cycle. Food consumption, water intake, body weight, and carcass composition were measured at weekly intervals. Body composition was determined using nuclear magnetic resonance (NMR) spectroscopy (Bruker Minispec, Billerica, MA, USA). After eight weeks on their respective diets, animals were transferred to the Metabolic Phenotyping Core at Pennington Biomedical Research Center for measurement of energy expenditure (EE) using a Promethion indirect calorimetry system (Sable Systems, Las Vegas, NV, USA). Mice were acclimated to the metabolic chambers for one week prior to data collection over four consecutive days. VCO_2_, VO_2,_ and activity measurements were collected at five-minute intervals over the 96 h period. Animals were then returned to their home cages to continue on their assigned diets for a two-week re-equilibration period. Thereafter, animals were fasted for four hours prior to euthanasia via CO_2_-induced narcosis and decapitation. Trunk blood was collected for serum analyses, and livers and inguinal white adipose tissue (IWAT) were harvested and snap frozen in liquid nitrogen and stored at −80 °C until analyzed.

### 2.2. Analysis of Energy Expenditure

VO_2_ is expressed as liters (L) of O_2_ consumed per h, while Respiratory Exchange Ratio (RER) is the ratio of VCO_2_ produced to VO_2_ consumed. EE was calculated as (VO_2_ × (3.815 + (1.232 × RER)) × 0.96 kCal/h) × 4.019 kJ/kCal, and expressed as kJ/h/mouse as described by the manufacturer (Promethion, Sable Systems, N Las Vegas, NV, USA). Group differences in 24 h EE (kJ/h/mouse) at study′s end were compared using Analysis of Covariance (ANCOVA) (JMP Software, Version 15; SAS Institute Inc., Cary, NC, USA) to calculate least squares means that accounted for variation in EE attributable to differences in lean mass, fat mass, food intake, and activity among the mice as before [18].

### 2.3. Serum Metabolite Analyses

Fasting serum FGF21 (R&D Systems; Minneapolis, MN, USA) was determined via enzyme-linked immunosorbent assays (ELISA) according to the manufacturer′s protocol.

### 2.4. RNA Isolation and qPCR of MR Target Genes

Total RNA was isolated from livers and IWAT from each mouse using a RNeasy Mini kit (Qiagen, Valencia, CA, USA). RNA concentration in each sample was measured using a Nanodrop ND-1000 spectrophotometer (Nanodrop Technologies, Wilmington, DE, USA). Total RNA was analyzed using the Agilent Bioanalyzer RNA 1000 chip (Agilent Technologies, Santa Clara, CA, USA) to confirm integrity of the RNA as indicated by RIN values > 7. Two μg total RNA was used for reverse transcription to produce cDNA. mRNA expression for previously identified MR target genes in liver [19,20] and IWAT [21,22] was measured using 10 ng cDNA via quantitative PCR using SYBR Green (Bio-Rad, Valencia, CA, USA), and mRNA concentrations of each target gene were standardized to cyclophilin expression. Primer sequences are provided in Appendix A.

### 2.5. RNAseq Analysis

Liver RNA samples were processed for library construction using the Lexogen Quant-Seq 3′ mRNA-Seq Library Prep Kit as previously described [23]. All libraries were pooled in equimolar amounts and sequenced on the Illumina NextSeq 500 at 75 bp forward and 6 bp forward index reads. Preliminary data analysis was performed using the Lexogen Quantseq pipeline 2.3.6 FWD on the Bluebee platform for quality control, mapping, and read count tables. The gene expression profiles were assessed using 6 replicates from each dietary group. The expression profiling data have been deposited in NCBI under GEO accession GSE168132.

### 2.6. Bioinformatics Analysis

Prior to differential gene expression analysis, scaled normalized count data for samples from the dietary groups (Cas-20%, Cas-5%, Soy-20% + SAA, Soy-10%) were analyzed via principal component analysis (PCA) (using *prcomp* package in R, http://www.R-project.org, accessed on 15 May 2021) to cluster samples based on gene expression similarities, and to identify potential outliers. Then, differential analysis of RNA read count data was performed using DESeq2 software [24], which models read counts as a negative binomial distribution and uses an empirical Bayes shrinkage-based method to estimate signal dispersion and fold changes. Gene expression signals were logarithmically transformed (to base 2) for all downstream analyses (the lowest expression value being set to 1 for this purpose). Genes with an absolute log fold change ≥ 1 and false discovery rate (FDR) of 5% were considered as differentially expressed.

*Ingenuity Pathway analysis (IPA)*—Pathway over-representation analysis was conducted using IPA (QIAGEN Inc., Valencia, CA, USA, https://www.qiagenbioinformatics.com/products/ingenuity-pathway-analysis, accessed on 15 May 2021), considering 2829 differentially expressed genes from the Cas-5% to Cas-20% samples and 2158 differentially expressed genes from the Soy-10% to Soy-20% + SAA samples (absolute log fold change ≥ 1.3, FDR ≤ 0.3). Within IPA, the Upstream Regulator Analysis module was utilized to identify putative gene regulators responsible for the observed transcriptional patterns produced by the respective low protein diets compared to the respective control diets for each protein source. Several possible types of upstream regulators were considered but special emphasis was placed on transcription factors, cytokine receptors, G protein-coupled receptors, and ligand-dependent nuclear receptors. Differential regulation of canonical pathways by the protein-restricted diets was also examined. Upstream regulators and canonical pathways with an activation z-score ≥ 2 or ≤−2 were considered to be activated or inhibited, respectively. Heat maps were used to visualize upstream regulators and canonical pathways that were differentially affected by the two low-protein diets relative to their respective control diets.

*Pre-ranked GSEA analysis*—Enrichment analysis of biological pathways (gene sets) was conducted via gene set enrichment analysis (GSEA) [25]. Specifically, the classical (unweighted) GSEA option was used with gene set permutation mode and enrichment computed on pathways present in the Kyoto Encyclopedia of Genes and Genomes (KEGG) database [26] available from the Molecular Signatures Database repository (MSigDb, http://software.broadinstitute.org/gsea/msigdb, accessed on 15 May 2021) [27]. Gene-sets with FDR ≤ 5% were considered as significantly enriched [28]. The individual contributions of pathway genes to the pathway enrichment signal were visualized via enrichment plots depicting the trajectory of a normalized pathway enrichment score against the rank of pathway genes in the context of the full gene list.

### 2.7. Data Analysis

Body weight, adiposity, food intake, water intake, serum FGF21, and expression of specific genes were analyzed using one-way ANOVA (GraphPad Prism; San Diego, CA, USA) with diet as the main effect, and residual variance used as the error term to calculate standard errors for comparisons of means across groups. Group differences in EE (kJ/hr/mouse) at the end of the study were compared using ANCOVA as described previously [18,29]. Protection against type I errors for all comparisons was set at 5% (α = 0.05).

## 3. Results

### 3.1. Effects of Protein Restriction on Energy Balance

Time-dependent changes in body weight and fat mass were compared in mice subjected to protein restriction using diets formulated from either casein or soy. Casein was restricted from 20% to 5% and soy was restricted from 20% to 10% to produce a comparable degree of sulfur amino acid (e.g., SAA (methionine and cystine)) restriction. Methionine and cystine concentrations were equalized between the casein control (e.g., Cas-20%) and soy control diets (Soy-20% + SAA) during formulation as illustrated in Table 1. The amino acid composition of the casein and soy protein sources used to formulate the diets is provided in Appendix A. Methionine and cystine were also equalized between the two protein-restricted groups. (e.g., Cas-5% and Soy-10%) so that an equivalent four-fold restriction of SAA concentrations were achieved in both the casein- and soy-based diets. The Soy-20% diet served as a negative control diet for the SAAs added to the soy-20% + SAA diet. Casein restriction slowed the accumulation of body weight (Figure 1A) and fat mass (Figure 1E) such that by three weeks, body weight and fat mass were significantly lower in the Cas-5% group compared to the Cas-20% group. A similar pattern was observed among the soy diet groups, with the Soy-10% group accumulating body weight (Figure 1B) and fat mass (Figure 1F) significantly slower than the Soy-20% + SAA and Soy-20% groups. The differences became significant after two weeks on the respective diets, and interestingly, the Soy-20% + SAA and Soy-20% groups did not differ at any time point. The three control groups are compared in Figure 1C,G and show that the Cas-20% group accumulated body weight and fat mass noticeably slower than the two soy control groups, although the differences did not become significant until the last 3–4 weeks of the study. However, when the change in body weight (Figure 1D) and fat mass (Figure 1H) were expressed relative to their respective control groups, it is evident that protein restriction implemented with either casein or soy produced comparable limiting effect on body weight and fat mass accumulation over time.

Average food (Figure 2A) and water intake (Figure 2B) over the course of the study were increased by both casein restriction and soy restriction. Casein restriction produced a larger effect on food intake than soy restriction (Figure 2A), but the effects of restriction of the respective protein sources on water intake did not differ (Figure 2B). Interestingly, food and water intake in the Soy-20% and Soy20% + SAA did not differ and were comparable to food and water intake in the Cas-20% group (Figure 2A,B). Casein restriction increased both nighttime and daytime EE measured at the end of the study, but the nighttime increase in the Cas-5% group compared to the Cas-20% group was much larger than the increase during the day (Figure 2C).

EE was also increased in the Soy-10% group compared to the two soy control groups, although the magnitude of the nighttime and daytime differences among the groups appeared smaller (Figure 2D). Support for this observation comes when the average EEs over the 4-day measurement period are compared (Figure 2E). The increase in EE produced by the Cas-5% group compared to its control is significantly larger than the increase in EE produced by the Soy-10% diet compared to its controls. Previous work has shown that the ability of protein restriction to increase EE is dependent on an increase in hepatic FGF21 [12], so it was not surprising that both casein restriction and soy restriction produced significant increases in hepatic *Fgf21* mRNA and serum FGF21 (Figure 2F). Although the increase in hepatic *Fgf21* mRNA was comparable between the two protein sources, casein restriction produced a significantly greater increase in serum FGF21 compared to soy restriction (Figure 2F). It is unclear whether this observation is related to the slightly smaller increase in EE produced by soy restriction compared to casein restriction (Figure 2E).

### 3.2. Transcriptional Effects of Protein Restriction in Liver and Adipose Tissue

To test whether the previously reported transcriptional effects of casein restriction on liver [12] and adipose tissue target gene expression [30] were recapitulated by soy protein restriction, representative ATF4 target genes and lipogenic genes were measured in the liver, and genes associated with adipose tissue remodeling were measured in inguinal WAT. Casein and soy protein restriction produced comparable significant increases in *Asns* and *Psat1* expression and decreases in *Fasn* and *Scd1* expression (Figure 2G). In inguinal WAT, expression of the nuclear-encoded mitochondrial genes, *Cox7a*, *Cox8b*, *Ucp1*, and *Cidea* were all significantly increased by casein and soy restriction (Figure 2H). However, the increase in *Cox7a* and *Ucp1* mRNA by Soy-10% was significantly less than the increases in these genes produced by Cas-5%. With the exception of a small but significant increase in *Ucp1* mRNA produced by Soy-20%, the mice fed the control casein and soy diets had comparable expression of these four genes (Figure 2H).

### 3.3. Differential Gene Expression in the Liver

To obtain a more comprehensive assessment of the transcriptional responses to casein and soy protein restriction, we used RNAseq in conjunction with bioinformatics analysis to interrogate the transcriptional responses in livers from mice fed the Cas-20%, Cas-5%, Soy-20% + SAA, and Soy-10% diets. The Soy-20% control group was excluded from this analysis because of its similarity to the Soy-20% + SAA group in essentially all responses. To assess the uniformity of differential gene expression across dietary protein sources, heat maps of the top 50 genes for each phenotype in input normalized counts were used to create Ranked Gene List Correlation Profiles for the Cas-5% to Cas-20% comparison (Appendix A) and the Soy-10% to Soy-20% + SAA comparison (Appendix A). These figures show the consistent uniformity of expression across samples within and across dietary group comparisons.

To further explore the systems biology of hepatic transcriptional responses to casein versus soy protein restriction, Ingenuity Pathway Analysis (IPA) was used to screen differentially expressed genes against Ingenuity′s annotated Knowledge Base to detect predicted activation or inhibition of canonical pathways and upstream regulators. This algorithm predicts changes in canonical pathways and transcription factor activity based on observed patterns of change in expression of genes known to be up- or down-regulated by specific upstream regulators. Heat maps were constructed to illustrate the comparative effects of casein restriction or soy restriction on the twenty-five most up- or down-regulated canonical pathways (Figure 3A) and upstream regulators (Figure 3B). When canonical pathways were evaluated in terms of activation or inhibition, fourteen pathways were comparably inhibited by both forms of protein restriction, while only the eIF2 Signaling pathway was comparably activated by casein and soy restriction (Figure 3A). In contrast, eight canonical pathways were activated by soy restriction but inhibited by casein restriction (Figure 3A). Viewed together, these findings show that despite substantial similarities in the way restriction of the two proteins affected upstream regulators and transcription factors in the liver (Figure 3B), about 30% of the canonical pathways were differentially regulated by casein restriction compared to soy restriction (Figure 3A).

To obtain a more detailed assessment of the differential effects of the two protein sources on the hepatic transcriptome, GSEA was used to identify pathways in the Kyoto Encyclopedia of Genes and Genomes (KEGG) database that were comparably or differentially regulated by the respective forms of protein restriction. For example, the GSEA enrichment plots and associated heat maps for the Ribosome pathway presented in Figure 4A–D; show that protein restriction produces a comprehensive activation of this pathway with both casein (Figure 4A,C) or soy restriction (Figure 4B,D). GSEA was also used to test for differential enrichment in a comparison of the respective control groups (e.g., 20% Casein and 20% Soy + SAA) and no differences were found (data not shown). GSEA enrichment plots and the associated heat maps for the Drug Metabolism Cytochrome P450 pathway show that casein (Figure 5A,C) or soy protein restriction (Figure 5B,D) produced significant but differential effects on gene set enrichment of this pathway. As with the Ribosome pathway, differential expression levels of the genes within this set were not detected between the 20% Casein and 20% Soy + SAA control groups. The effects of casein and soy restriction on the Arginine Proline Metabolism gene set were similar to their effects on the Ribosome and Drug Metabolism pathways in the sense that differential expression of genes within the Arginine Proline gene set was not detected between the 20% Cas and 20% Soy + SAA control groups.

However, restriction of each protein source produced a similar degree of enrichment of genes in this set that were downregulated by 5% Cas and 10% Soy (Appendix A). Significant gene set enrichment was also detected for seven other amino acid metabolism pathways when casein or soy protein were restricted, but there were subtle but significant differences in the way the gene sets were affected by the two protein sources. For example, with the Alanine Aspartate Glutamate Metabolism pathway, the gene set enrichment score indicated that a subset of genes in this pathway were expressed at higher levels in the 20% Cas compared to 20% Soy + SAA controls. Thus, although restriction of casein or soy produced comparable gene set enrichment of genes downregulated by restriction of each protein compared to controls (Appendix A), the 5% Cas diet produced a stronger downregulation of this gene set. In contrast, the Tyrosine Metabolism, Hisitidine Metabolism, and Glycine Serine Threonine Metabolism gene sets were expressed at comparable levels between the 20% Cas and 20% Soy + SAA control groups. However, the enrichment scores for genes downregulated by protein restriction within each pathway indicated that casein restriction produced a significantly stronger downregulation of these gene sets than soy restriction for all three pathways (Appendix A). It is also worth noting that soy protein restriction produced a slightly larger upregulation of a small subset of genes in each pathway compared to casein restriction (Appendix A). The effects of protein restriction on the Cysteine Methionine Metabolism pathway were unique in that a notable subset of genes within this pathway were expressed at higher levels in the 20% Cas compared to the 20% Soy + SAA group (data not shown). This difference in starting points resulted in casein restriction producing a stronger downregulation of a subset of genes compared to soy restriction, and a stronger upregulation of a small subset of pathway genes by soy restriction compared to casein restriction (Appendix A). Lastly, the effects of protein restriction on the Valine Isoleucine Leucine Degradation and Trytophan Metabolism pathways were similar and based on a significantly stronger downregulation of a large subset of genes in both pathways by casein restriction compared to soy restriction (Appendix A). Collectively, these findings make a compelling case that implementation of protein restriction using diets formulated from different protein sources can have both differential and common effects across multiple pathways involved in hepatic amino acid metabolism. Paradoxically, these findings also show that despite these differences, comparable degrees of sulfur amino acid restriction using different protein sources can produce a number of common beneficial biologic effects bridging multiple physiological, biochemical, and metabolic pathways. We conclude that methionine restriction can be implemented using dietary restriction of different sources of protein.

## 4. Discussion

Dietary protein restriction and dietary methionine restriction produce a comparable set of behavioral, physiological, biochemical, and transcriptional effects in the weeks and months after their introduction [11,12,13,17,29,30,31,32]. Both experimental diets increase food and water intake, increase EE, limit fat deposition, increase hepatic transcription and release of FGF21, decrease the capacity of the liver to conduct *de novo* lipogenesis, decrease hepatic and circulating lipid levels, increase tissue-specific and overall insulin sensitivity, and remodel white and brown adipose tissue to more thermogenically active states (for reviews see [2,4]). The cumulative effect of these responses is to improve metabolic health, enhance healthy aging and ultimately to increase longevity of the animal [32,33,34,35,36]. In mice, the prototypical model of protein restriction involves restriction of casein from 20% to 5%, effectively reducing dietary sulfur amino acid concentrations from 0.57% methionine and 0.14% cystine to 0.125% methionine and 0.035% cystine. However, implementation of casein protein restriction typically involves supplementing the 5% casein diet with 0.075% cystine, bringing the final sulfur amino acid concentration to 0.235% [11]. In the most widely used implementation of dietary methionine restriction, the control and methionine-restricted diets are formulated from elemental amino acids to contain 0.86% methionine in the control diet and 0.17% methionine in the methionine-restricted diet [37,38]. Cystine is not included in either diet, forcing the animal to use dietary methionine and catabolism of tissue proteins to obtain sufficient cysteine for synthesis of taurine and glutathione [39]. Given that casein restriction to 5% faithfully reproduces the full metabolic profile of dietary methionine restriction and restricts sulfur amino acids to a similar extent, it seems likely that dietary casein restriction is producing some if not all of its biological effects by limiting dietary methionine intake. If this hypothesis is correct, one should be able to reproduce the biological effects of casein restriction using a different protein source by carefully formulating control and experimental diets to match the sulfur amino acid concentrations to the 20% and 5% casein diets. Using these parameters, Research Diets, Inc. created two 20% soy control diets, one with no added methionine or cystine and a second with methionine and cystine added to match the concentrations of methionine and cystine present in the 20% casein control diet. Additionally, by restricting soy to 10%, the final concentrations of methionine and cystine were closely matched to the concentrations of methionine and cystine in the 5% casein diet.

The physiological, endocrine, and molecular responses to the five diets are summarized in Figure 1 and Figure 2, and reveal several unanticipated findings. First, both soy control diets (e.g., 20% Soy and 20% Soy + SAA) supported a faster rate of growth and fat deposition than the 20% Casein control diet (Figure 1C,G). The reason for these differences is unclear, although a previous short-term study also observed faster growth in mice fed a soy-based compared to casein-based diet [40]. In an experiment of similar design, Zeng et al. detected a small, but significantly higher rate of food intake of a soy-based control diet compared to the casein diet, but detected no effect on body weight [41]. In the present work, food intake did not differ between the 20% casein and 20% soy diets (Figure 2A). Second, in all other respects, the responses to the two soy control diets were quite similar to the responses observed in the 20% Casein control group (Figure 2A–H). Third, when the differences in growth between the soy control groups and casein control group were accounted for, the body weight and fat deposition responses to restriction of casein or soy were essentially the same (Figure 1D,H). Fourth, the soy-restricted diet produced a significant increase in food intake (Figure 2A), water intake (Figure 2B), EE (Figure 2D,E), and serum FGF21 (Figure 2F), but in each case the magnitude of the effects was slightly but significantly less robust than the effects produced by casein restriction (Figure 2A,C–F). The larger effect on EE by casein restriction may be linked to the larger increase in serum FGF21 produced by casein compared to soy. However, restriction of the two proteins produced a comparable increase in hepatic *Fgf21* mRNA (Figure 2F), four other hepatic target genes (Figure 2G), and two of four genes in inguinal WAT (Figure 2H). Even′s lab reported many of the same effects in female Balb/c mice fed casein- and soy-restricted diets [42,43], but neither of their 6% low protein diets had any effect on body weight or adiposity [42,43]. This differs from the present findings and their findings may be due to their use of female mice. Direct comparisons of the effects of methionine restriction in male and female mice showed that young females on the MR diet preserved adipose tissue at the expense of lean, whereas male mice preserve lean mass at the expense of fat [21]. Blais et al. did observe a decrease in lean body mass of their protein-restricted female mice studied at 22 °C [42]. Viewed collectively, the overall similarity in responses produced by casein and soy restriction observed here argue that the equivalent degrees of sulfur amino acid restriction produced by the diets is an important determinant of these comparable biological effects.

A more comprehensive evaluation of the transcriptional impact of protein restriction using casein versus soy was obtained using high content analysis of the hepatic transcriptome. Using Ingenuity Pathway Analysis, the top 25 canonical pathways and upstream regulators/transcription factors were identified as being activated or inhibited by restriction of casein or soy relative to their corresponding control groups. The algorithm predicts activation or inhibition of canonical pathways or upstream regulators based on observed patterns of change in genes known to be components of those pathways. It is interesting that transcriptional programs regulated by methionine restriction in multiple tissues [17,44,45,46], are also regulated by protein restriction in the current study. These findings support the view that hepatic methionine sensing plays an important role as a mediating mechanism of the observed transcriptional responses to protein restriction. However, ~30% of the canonical pathways were also differentially regulated by casein compared to soy restriction (Figure 3A). This finding argues that unique differences between the two proteins are being sensed and produce opposite effects in these pathways. Dietary soy has long been recognized as producing beneficial metabolic responses that were independent of any effect of the protein on body size or composition. For example, in short-term studies with casein-based and soy-based control diets in Wistar or Sprague Dawley rats, the soy-based diet reduced hepatic expression of lipid synthesis genes, liver and circulating triglycerides, and fasting insulin [47,48]. Additionally, in long term studies comparing the two diets, insulin-dependent glucose disposal rates were increased in the soy-fed Wistar rats [48]. Some of these protein-specific responses may also be strain-specific, as shown in longevity studies where Fisher 344 rats consuming casein diets died earlier and at significantly higher rates from chronic nephropathy than rats on the soy-based diet [49]. Rats on the soy-based diets were mostly unaffected by this single cause of death [49]. Considered together, these studies show that even matched casein and soy diets can produce notably different metabolic responses. However, in the present work the two forms of protein restriction produced a large number of highly comparable transcriptional responses in the liver. For example, comparison of the activated and inhibited upstream regulators supports the view that casein and soy restriction produced comparable signaling inputs to a large number of molecules that regulate transcriptional programs in the liver (Figure 3B). Figure 3B shows that the directionality of the effects of casein restriction and soy restriction were similar in all cases, differing only in the degree of activation or inhibition.

Gene set enrichment analysis was also employed to look for common or differential effects of the two types of protein restriction on hepatic gene sets. The Ribosome pathway, indicative of changes in RNA biology associated with protein translation, was the highest scoring pathway for both casein restriction and soy restriction. The enrichment plots (Figure 4A,B) and corresponding heat maps make a compelling case that casein and soy restriction were producing comparable transcriptional effects on RNA biology in the liver. This conclusion is supported by the strong, comparable activation of eIF2 signaling (see Figure 3A) that is known to be activated by essential amino acid restriction [20,23]. Perhaps the largest overall impact of restriction of the two proteins was on hepatic amino acid metabolism, as significant enrichment was detected in the Arginine Proline metabolism pathway, the Alanine Aspartate metabolism pathway, the Tyrosine metabolism pathway, the Histidine metabolism pathway, the Glycine Serine Threonine metabolism pathway, the Cysteine Methionine metabolism pathway, the Valine Isoleucine Leucine degradation pathway, and the Tryptophan metabolism pathway. These differences may be reflective of the differences in methionine, cysteine, alanine, aspartate, and glutamate content of the two proteins, while most other amino acids are similar between casein and soy (Appendix A). It seems probable that these differences are reflective of the previously unrecognized complexity of the liver′s responses to restriction of proteins with differing amino acid composition. However, despite the complexity of the differing responses to restriction of casein versus restriction of soy in terms of amino acid metabolism pathways (Appendix A), the physiological and metabolic responses to casein restriction and soy restriction are mostly comparable and seemingly unaffected by these differences.

An important implication of the present work is that it supports the feasibility of deriving the therapeutic benefits of dietary methionine restriction by restricting protein intake in conjunction with a careful accounting of methionine content of the various proteins that make up the diet. The most straightforward way to restrict methionine is to formulate amino acid-based diets that reduce the methionine content to ~0.17% and eliminate cystine. This is the approach used in pre-clinical studies of methionine restriction and rodents readily adapt to consumption of these diets. However, this approach works poorly in humans because of poor tolerance of the bitter, metallic taste of elemental amino acids. The medical food, Hominex^®^-2 is a methionine free mixture of essential amino acids that was developed to provide nutritional support to patients with pyridoxine-insensitive hypercystinuria or hypermethionemia [50]. Although Hominex-2 was moderately effective in increasing fat oxidation and reducing hepatic lipid content in patients with metabolic syndrome, the high withdrawal rates from the study and subsequent feedback made it clear that poor palatability was a significant drawback [51]. A second major drawback to the use of Hominex-2 is that it contains significant amounts of cystine, which effectively spares methionine and lessens the severity of the methionine restriction [39]. As shown by several authors, the addition of even small amounts of cystine to methionine-restricted diets effectively reverses the metabolic effects of methionine restriction [52,53,54]. For example, in work using diets containing 0.17% methionine, the addition of 0.2% cystine completely reversed the ability of the 0.17% methionine diet to increase EE and reduce fat mass [54]. Therefore, a critical question to be answered is how much dietary cystine can be present for a given amount of methionine restriction and still preserve the effects of the methionine restriction alone. In earlier work, the upper threshold of methionine restriction was ~0.25% methionine when no cystine was present in the diet [8]. In the current work, the sum of methionine and cystine together in the two protein-restricted diets was in the range of 0.24% to 0.26%. Together these findings and the present work illustrate that careful attention to the total amount of methionine and cystine will be needed to successfully implement a protein-restricted diet that produces therapeutically effective reductions in sulfur amino acids.

Lastly, an alternative approach to dietary methionine restriction was recently described which reduced the methionine and cystine content of casein by targeted, oxidative deletion of the sulfur amino acids in the intact protein [19]. The advantage of this approach is that the sulfur amino acid-depleted casein maintains its palatability and can produce the metabolic benefits of protein restriction without reducing overall protein content of the diet. Proof of concept studies comparing methionine-depleted casein-based diets to elemental amino acid-based methionine restricted diets established the feasibility of this approach and showed that the beneficial metabolic effects of methionine restriction were reproduced by the oxidized casein-based MR diets [19]. It will be interesting in future studies to determine whether combinations of mild protein restriction coupled with targeted methionine depletion of the proteins to be restricted can be implemented to produce therapeutically effective diets for the treatment of obesity and metabolic disease.

## 5. Conclusions

The present work establishes that restriction of dietary soy protein content from 20% to 10% is able to fully recapitulate the biological effects of restriction of dietary casein from 20% to 5%, and that the efficacy of both protein-restricted diets is linked to their comparable restriction of dietary sulfur amino acids. These findings support the feasibility of using different protein sources to formulate therapeutically effective protein-restricted diets that limit methionine and cysteine intake to levels that reproduce the biological effects of dietary methionine restriction.

## Figures and Tables

**Figure 1 nutrients-13-02609-f001:**
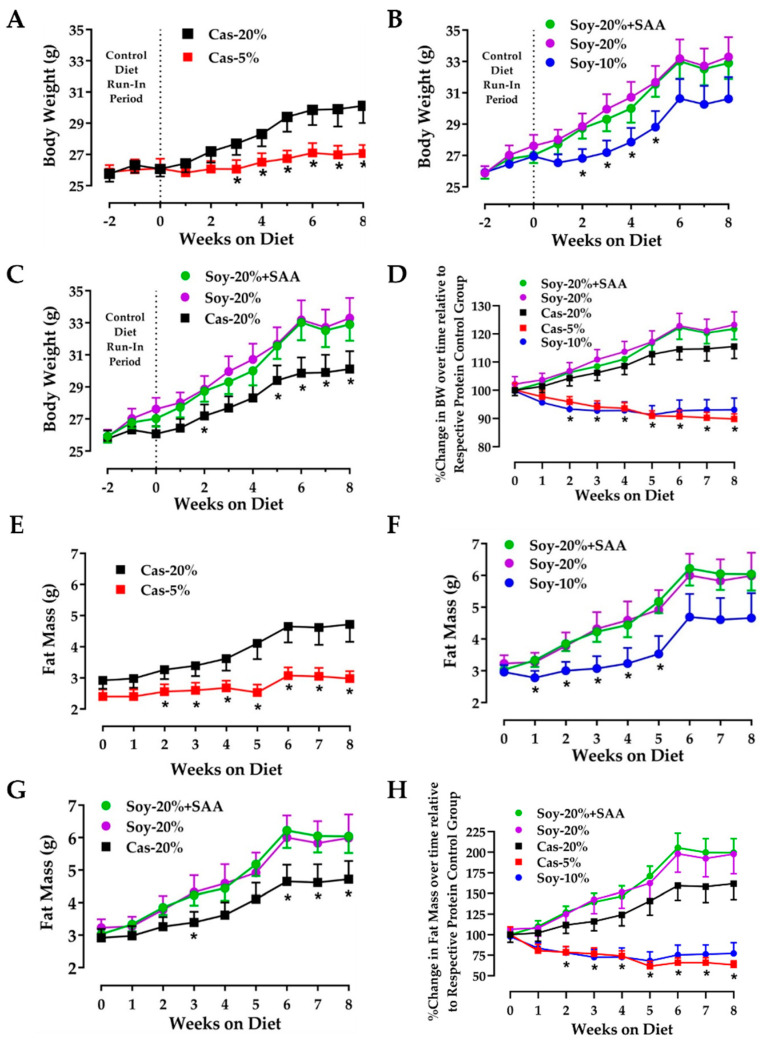
Impact of protein restriction on body weight (**A**–**D**) and fat mass (**E**–**H**) in mice fed control or protein-restricted diets for 8 weeks beginning at 12 weeks of age. Cas-20% served as the control group for casein protein restriction and Soy-20% and Soy-20% + SAA served as control groups for soy protein restriction. Dietary composition of the respective diets is described in Materials and Methods. Body weight and composition were measured weekly for the entire study. In Figure 1D,H, the changes in body weight (Figure 1D) and fat mass (Figure 1H) were expressed as percent change in the respective variable relative to the body weight and fat mass at zero time. The change in body weight and fat mass over time were analyzed using a repeated measures one-way ANOVA as described in the Materials and Methods, and means from protein-restricted groups annotated with an ‘asterisk” differ from their respective control groups at *p* < 0.05. Data in each figure panel are presented as the mean ± SEM, *n* = 9–10.

**Figure 2 nutrients-13-02609-f002:**
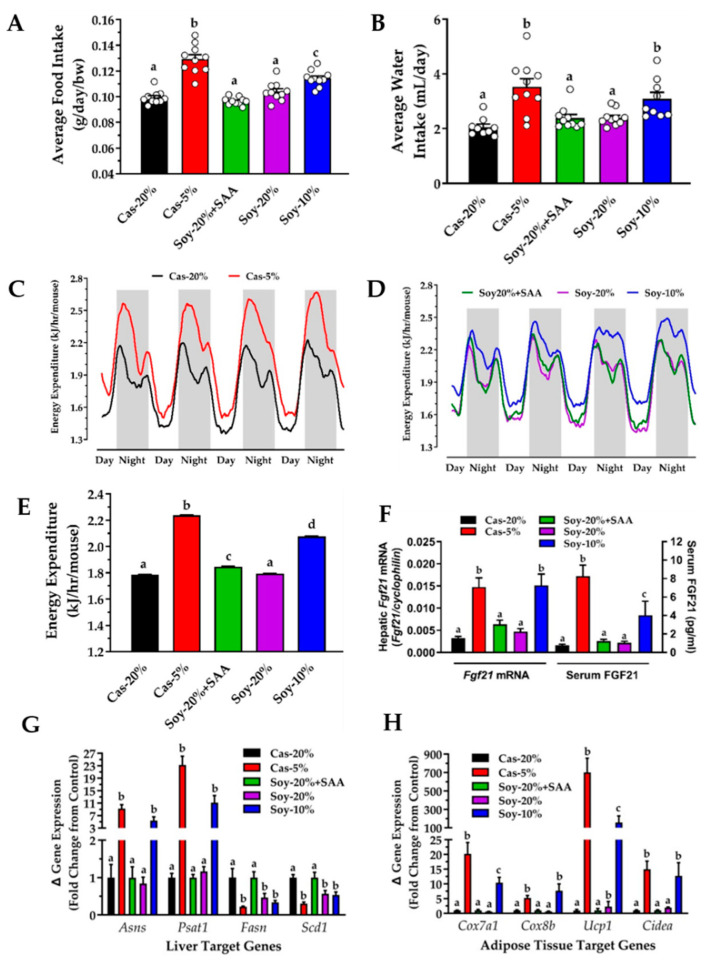
Assessment of impact of protein restriction on average food intake (**A**), average water intake (**B**), 96 h continuous energy expenditure in casein groups (**C**) and soy groups (**D**), average energy expenditure for all groups (**E**), *Fgf21* mRNA and serum FGF21 (**F**), liver target genes (**G**), and inguinal white adipose tissue target genes (**H**). Food intake and water intake were measured at weekly intervals, and average food intake was expressed per unit body weight. The means of food intake and water intake were averaged over the entire study. Energy expenditure was measured continuously for 4 days after a 3-day equilibration period and calculated by ANCOVA as described in Materials and Methods. Serum and liver collected at end of study was used to measure FGF21 and target gene mRNA levels. Inguinal WAT collected at the same time was used to measure target gene mRNA levels. The respective end points were analyzed by one-way ANOVA and means annotated with different letters differ at *p* < 0.05. Data in each figure panel are presented as the mean ± SEM, *n* = 9–10.

**Figure 3 nutrients-13-02609-f003:**
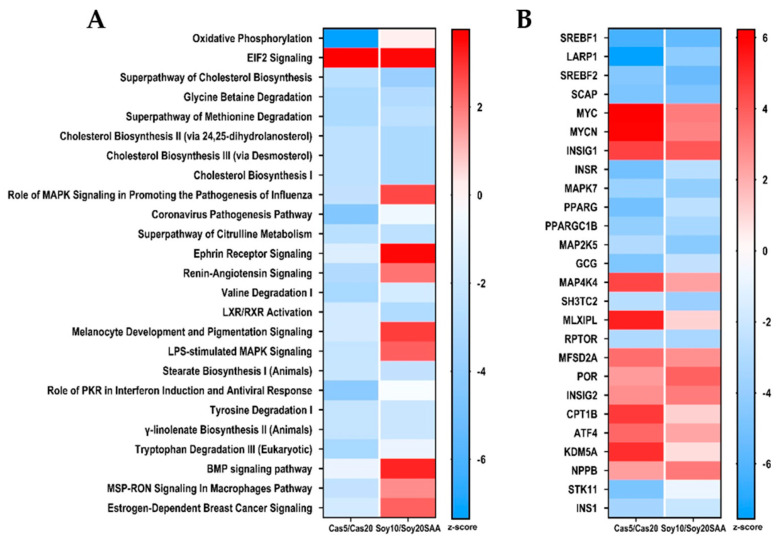
Bioinformatics analysis of hepatic gene expression in mice fed sulfur amino acid-matched control diets formulated from casein (Cas-20%) or soy (Soy-20% + SAA), and sulfur amino acid-matched protein restricted diets formulated from casein (Cas-5%) or soy (Soy-10%). Canonical pathway analysis (**A**) and upstream regulator analysis (**B**) were conducted as described in the Materials and Methods. Pathway analysis was conducted using livers from 6 mice of each diet and utilized to identify putative gene regulators responsible for the observed transcriptional patterns produced by the respective protein-restricted diets. Upstream regulators and canonical pathways with an activation z-score ≥ 2 or ≤ −2 were considered to be activated (red) or inhibited (blue), respectively. Heat maps were used to visualize the top 25 canonical pathways and upstream regulators that were differentially affected by the MR diet in the two genotypes.

**Figure 4 nutrients-13-02609-f004:**
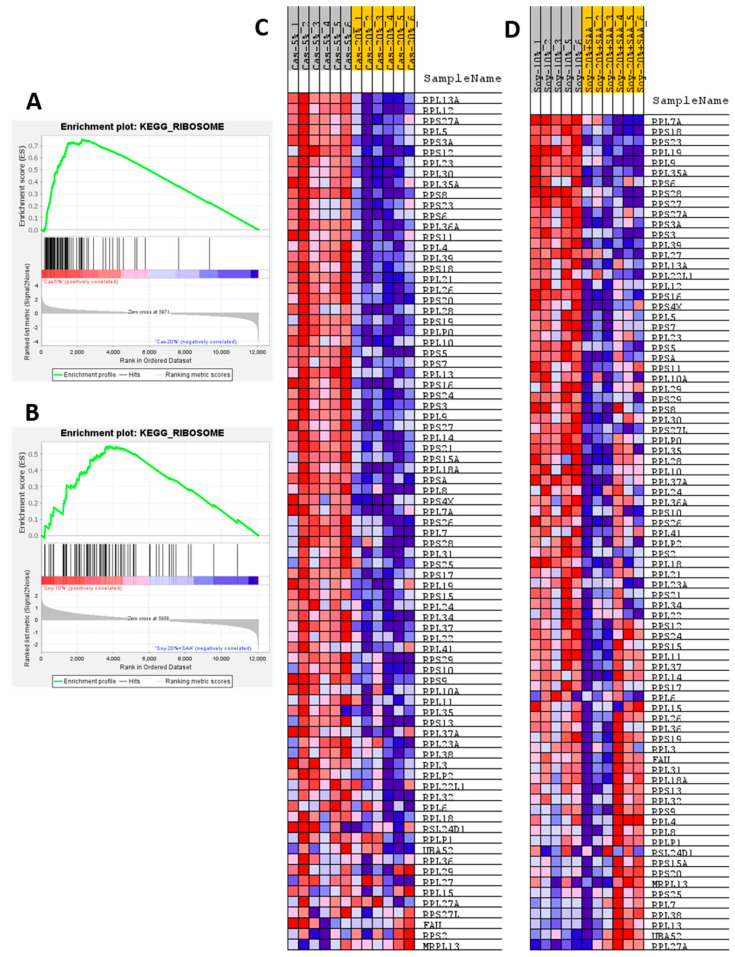
Gene set enrichment analysis (GSEA) of hepatic gene expression in mice fed sulfur amino acid-matched control diets formulated from casein (Cas-20%) or soy (Soy-20% + SAA), and sulfur amino acid-matched protein restricted diets formulated from casein (Cas-5%) or soy (Soy-10%). The enrichment plots and corresponding heat maps are shown for the top-scoring KEGG Ribosome gene set for the Cas-5% to Cas-20% (**A**,**C**) and Soy-10% to Soy-20% + SAA comparisons (**B**,**D**). Enrichment was computed as described in the Material and Methods and enrichment plots were generated to visualize the individual contributions of pathway genes to pathway enrichment. Accompanying heat maps present the normalized enrichment scores for individual genes within the gene set [blue = downregulation, red = upregulation, gray = not significant (FDR > 0.1)].

**Figure 5 nutrients-13-02609-f005:**
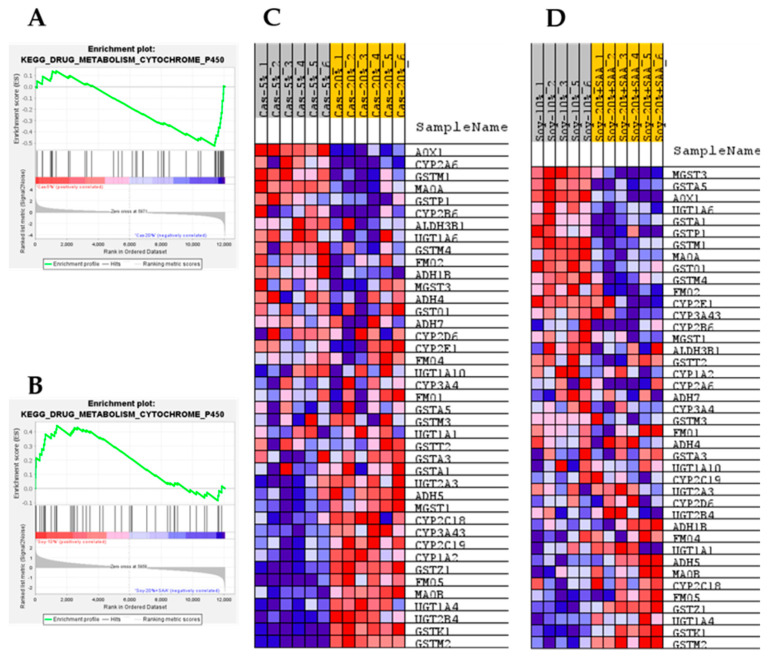
Gene set enrichment analysis (GSEA) of hepatic gene expression in mice fed sulfur amino acid-matched control diets formulated from casein (Cas-20%) or soy (Soy-20% + SAA), and sulfur amino acid-matched protein restricted diets formulated from casein (Cas-5%) or soy (Soy-10%). The enrichment plots and corresponding heat maps are shown for the KEGG Drug Metabolism Cytochrome P450 gene set for the Cas-5% to Cas-20% (**A**,**C**) and Soy-10% to Soy-20% + SAA comparisons (**B**,**D**). Enrichment was computed as described in the Material and Methods, and enrichment plots were generated to visualize the individual contributions of pathway genes to pathway enrichment. Accompanying heat maps present the normalized enrichment scores for individual genes within the gene set [blue = downregulation, red = upregulation, gray = not significant (FDR > 0.1)].

**Table 1 nutrients-13-02609-t001:** Composition of Control and Protein-restricted diets formulated from casein or soy protein isolates to produce matching concentrations of methionine and cystine in both casein-based and soy-based control and protein-restricted diets.

	Cas-20%	Cas-5%	Soy-20% ^1^	Soy-20% + SAA	Soy-10%
Casein or soy protein, g	200	50	200	200	115
Grams of added Met	0	0	0	3.53 ^2^	0
Final [Met], gm per 100 g diet	0.57	0.14	0.22	0.57	0.13
Grams of added Cys	3.00 ^3^	0.75 ^4^	0	1.91 ^5^	0
Final [Cys], gm per 100 g diet	0.41	0.10	0.22	0.40	0.13
Corn Starch, g	376	485	376	370	426
Maltodextrin 10, g	125	150	125	125	150
Sucrose, g	107	107	107	107	107
Cellulose, g	50	50	50	50	50
Soybean Oil, g	25	25	25	25	25
Lard, g	75	75	69.6	69.6	72.3
Mineral Mix S10022C, g	3.5	3.5	3.5	3.5	3.5
Calcium Carbonate, g	12.5	8.7	10.6	10.6	9.3
Calcium Phosphate, g	0	5.3	0	0	3.0
Potassium Citrate, 1 H_2_O, g	2.5	2.5	2.5	2.5	2.5
Potassium Phosphate, g	6.9	6.9	6.9	6.9	6.9
Sodium Chloride, g	2.6	2.6	2.6	2.6	2.6
Vitamin Mix V10037, g	10	10	10	10	10
Choline Bitartrate, g	2.5	2.5	2.5	2.5	2.5
**Total**	1001	985	991	991	986

^1^ The Soy-20% diet served as a negative control group with no added sulfur amino acids and only naturally occurring amounts of methionine and cystine in the diet. ^2^ 3.5 g of Methionine was added to each kg of the Soy-20% + SAA diet to bring the final methionine concentration to 0.57% to match the amount of methionine in the Cas-20% diet. ^3^ 3.0 g of Cystine was added to each kg of the Cas-20% diet to bring the final cystine concentration to 0.4% to match the amount of cystine in the Soy-20% + SAA diet. ^4^ 0.75 g of Cystine was added to each kg of the Cas-5% diet to bring the final cystine concentration to 0.1% to closely match the amount of cystine in the Soy-10% diet. ^5^ 1.91 g of Cystine was added to each kg of the Soy-20% + SAA diet to bring the final cystine concentration to 0.4% to match the amount of cystine in the Cas-20% diet.

## Data Availability

All data generated or analyzed during this study will be available from the lead contact upon request.

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
