# Peer review of "The Role of Reduced Methionine in Mediating the Metabolic Responses to Protein Restriction Using Different Sources of Protein"

_nutrients, 2021, doi:10.3390/nu13082609_

Round 1

Reviewer 1 Report

The manuscript of Fang et al. compares efficacy of methionine-restricted diets formulated from either animal-based (casein) or plant-based (soy protein) protein source to recapitulate the metabolic phenotype produced by methionine restriction using elemental diets. Authors conclude that restriction of dietary soy protein fully recapitulates the biological effects of restriction of dietary casein and that this efficacy of both restrictive diets is linked to a reduced intake of dietary sulfur amino acids. Manuscript is relatively straightforward and easy to follow. I have the following comments for the authors to address in the revised version:

  • After reading the manuscript my first question is quite obvious. Why did you NOT include a group of mice on methionine restriction using elemental diets (& appropriate control group, if applicable) especially when the aim of the study was to compare efficacy of dietary protein restriction from various protein sources? I understand that it might have been performed by you or others several times before, but side by side comparison would be invaluable. Alternatively, if you previously published data on a similar study using same animal study design, you could have just included those data for comparison and refer to that paper. If needed, some analyses could have been performed on (e.g. RT-PCR, RNA-Seq), but that should not be a problem, as I am sure you would keep these samples in storage for eventualities like this one.
  • Introduction is sufficient, but I would ask author to add another short paragraph stating in one or two sentences the aim of the study/manuscript. It is mentioned in the abstract but needs to be at the end of Introduction as well.
  • Table 1 and elsewhere: What “gm%” stands for? I do not recognize this unit. It is not standard and not defined in the manuscript.
  • Supplementary material was missing as it was not part of the manuscript. However, legends for supplementary material are mentioned in the body of manuscript. I believe this is inappropriate and should be removed. A brief description of the content of the supplementary material with a reference where it can be accessed is appropriate and sufficient. After inquiring editorial office, I swiftly received pdf file with the supplementary material for review.
  • Authors present a ton of RNA-Seq expression data comparing differential expression of hepatic genes in response to given dietary interventions. This is great as an overview and overwhelms readers with data, but I entirely miss any in-depth analysis. Since the aim of the manuscript was comparison of protein restriction of various protein sources to methionine-restricted elemental diet, I would at least pay a special attention of genes involved in sulfur amino acid metabolism. Those are shown in Supplemetary figure 7, but to my surprise I did not find any mention of the key regulator of sulfur amino acid metabolism cystathionine beta-synthase (CBS). CBS is responsible for preserving sulfur in methionine cycle or its irreversible diversion through transsulfuration pathway to synthesis of cysteine. With those interventions I would expect significant changes in CBS expression. Do you have those data? If yes, how does it compare to CBS expression in mice on unrestricted diets as well as on Met-restricted elemental diet?
  • Authors mention cysteine in the content of the diets as well as for supplementation purposes. However, most diet-related literature refers to this amino acid in its oxidized, disulfide form cystine. Which one did you determine in your analyses? Is it really cysteine or indeed cystine? Please double check. Also in the Discussion, you should correct 0.24% to 0.235% as that is the number you get by adding 0.125% Met + 0.035% Cys + 0.075% supplemental Cys.
  • One potentially important drawback comparing the restricted diets from various protein sources with the effect of Met-restricted elemental diet (MR) is presence of Cys. You mentioned this briefly in the discussion, but it begs the question why you did not include MR diet supplemented with Cys to match in content those protein restricted diets from various protein sources?
  • Hominex-2 is used as diet-based treatment for pyridoxine-unresponsive homocystinuria. Please correct it.
  • References are numbered in the body of manuscript, but lacks numbering in the References section. Please correct it.

Reviewer 2 Report

The manuscript present an interesting study, which is of interest for the scientific community. However,the introduction section of the manuscript requires extensive revision. 

First of all, the authors need to expand the review of literature that is relevant to their study. 
Some examples: 
"Studies of the short-term metabolic effects of dietary protein restriction and methionine restriction (MR) have been pursued in parallel in recentyears, withstudies con-ducted in the last decade identifying significant similarities among the responses to the two dietary regimens. " - not supported by references. 
"Collectively, these studies illustrate that restricting dietary methionine to a specific range is highly beneficial, while a more severe restriction can be detrimental, and a less severe restriction is without effect. " - which studies?! no reference for "these studies" .

Second, and probably most importantly, the aim of the study needs to be properly highlighted and justified. Actually, the aim of this study is completely missing.... 
The other reviewer has been selected as well low for originality/novelty, significance of content and interest to the readers, compared to average of mine in evaluation. Therefore, the authors mustjust explain a bit more the novelty of this review and why now. I would suggest that the authors attempt to present the key objectives of their study with regards to what is currently known (i.e. literature), thus highlighting the added value of the paper, as originality/novelty, significance of content and interest to the readers has been evaluated as low to average. 

Third, the supplementary material was completely missing. 

Forth, conclusions are not presented properly, does not reflect and support the discussions. Please elaborate a bit this section.  
